# Do the Emotions of Middle-Income Mothers Affect Fetal Development More Than Those of High-Income Mothers?—The Association between Maternal Emotion and Fetal Development

**DOI:** 10.3390/ijerph16112065

**Published:** 2019-06-11

**Authors:** Dasom Kim, Insook Lee, Kyung-Sook Bang, Sungjae Kim, Yunjeong Yi

**Affiliations:** 1The Research Institute of Nursing Science, College of Nursing, Seoul National University, Seoul 03080, Korea; dudurdaram@naver.com (D.K.); ksbang@snu.ac.kr (K.-S.B.); sungjae@snu.ac.kr (S.K.); 2Department of Nursing, Kyung-In Women’s University, Incheon 21041, Korea; yinyis@hanmail.net

**Keywords:** maternal-fetal relations, emotions, psychological stress, fetal development, pregnancy

## Abstract

This study examines the relationship between the emotions of mothers and fetal development and explores the modifying effect that family income has on this relationship. Socio-demographic information, maternal depression, stress, positive and negative emotions, and maternal-fetal attachment data were collected at 16–20 weeks of pregnancy. Data on fetal body weight and biparietal diameter indicating fetal development were collected at 33–35 weeks to observe the longitudinal effects of mothers’ emotions on fetal development. We divided subjects into two groups: those with more than 150% of the median income were classified as the high-income group and less than 150% as the middle-income group. *T*-test, correlation analysis, and multiple regression analysis on maternal emotional status and fetal development were performed for each group. A positive correlation was found between maternal-fetal attachment and negative emotion that was associated with the biparietal diameter and fetal body weight only in the middle-income group. Results of the multiple regression analysis were statistically significant, indicating that maternal-fetal attachment was associated with fetal weight. These results show that the management of subjective emotion is associated with healthy development of the fetus and contributes to health equity.

## 1. Introduction

According to Jean-Jacques Rousseau’s theory of the origin of human inequality, over the years, humans have been unable to distinguish between their original nature and the one changed by society. Human beings experience two inequalities: genetic inequality and socially induced inequality. Inequality is caused by the notion of ownership and the desire for self-preservation [1]. As society develops, humans in unequal structures do not know how these inequalities may have affected them, and it is difficult to comprehend how lifestyles, eating habits, and the various emotions they experience affect their health. From this perspective, this study aims to identify where inequality starts.

We explored the relationship between socioeconomic and health inequities so we could use it to understand the actual cause of health differences and apply our insights to nursing interventions and policies. Social class structures may be a direct cause of the health outcomes that we are interested in, or an indirect variable that explains the causal relationship. Nursing practice should apply the principle of justice for the health of the entire population, and the influence of socioeconomic factors cannot be overlooked. In this study, we investigated the difference in maternal mental health, maternal-fetal attachment (MFA), and fetal development according to social class to improve the understanding of the health differences of pregnant women at different income levels.

There have been several investigations into how the prevalence of depression, anxiety, and stress vary according to socioeconomic level and affect both, the mother and child [2,3]. Factors affecting mothers’ mental health during pregnancy are as follows: individual factors, personal experiences, pregnancy-related factors, relationship factors, social conditions, and material conditions including poverty, lack of employment, and unstable housing [4]. In addition, in one path analysis, household income of pregnant mothers was the most influential indirect factor affecting maternal mental health [5]. Indeed, low-income countries are more likely to have higher rates of maternal depression and anxiety than are high-income countries [6].

Although the number of articles related to the emotional state of pregnant women during and after pregnancy has been increasing since 2000 [7], there is still very little scientific understanding of how pregnant women’s emotional states affect actual fetal development. It has been reported that high cortisol levels and depressed and anxious states slow the intrauterine growth and MFA [8]. Adequate maternal weight gain during pregnancy [9] is associated with intrauterine growth. However, studies on the relationship between fetal development and MFA are hard to find. One previous study has shown that MFA negatively affects fetal development in early, mid, and late pregnancies, but it did not explain the mechanism underlying the results [8].

There are two primary aims of this study: (1) To investigate whether a mother’s emotional state varies according to socioeconomic level; (2) To study longitudinal effects of how a mother’s emotions are associated with fetal development by repeated measures according to income level. Understanding whether there are emotional inequalities due to socioeconomic factors and the relationship between maternal emotion and fetal development will provide new knowledge for developing adequate prenatal care.

## 2. Materials and Methods

### 2.1. Research Objectives

This study used a longitudinal, prospective cohort design to examine the association between maternal emotions and fetal development with subgroup analysis by household income level.

### 2.2. Study Sample

A total of 161 pregnant women between 16–20 weeks of pregnancy were recruited from July 2017 to December 2018 and were followed up until 33–35 weeks of pregnancy. A total of 161 pregnant women were recruited. After excluding 40 dropouts (25%), 121 participants’ data were used in the final analysis. Twelve patients were transferred to other hospitals, two underwent abortion, and the others refused to participate in the study. The participants were recruited at a maternity hospital in Jeonju, South Korea. A research assistant with a registered nurse license checked the medical history and hospital records of the participants who met the inclusion criteria. The inclusion criteria were: (1) agreeing to participate in the study; (2) being primiparous; (3) having no underlying diseases (hypertension, diabetes, heart disease, thyroid disease); and (4) being pregnant with no experience of depression and anxiety. In addition, the exclusion criteria eliminated those with: (1) high-risk pregnancy (including placenta previa, preeclampsia, gestational diabetes, and high spontaneous abortion risk); (2) preterm birth (childbirth before 37 weeks of pregnancy), stillbirth, and abortion.

### 2.3. Procedure

This study was approved by Seoul National University institutional review board (IRB No. 1707/003-004). Prior to data collection, we received informed consents from the participants, and they were advised that they could drop out of the study at any time. Data were measured twice at 16–20 weeks (T1) and 33–35 weeks (T2) of gestation. The collected data were stored with a random number for anonymity and privacy, except from the responsible researchers and research assistants. Socio-demographic information, maternal depression, maternal stress, positive and negative emotion status, and MFA data were collected at 16–20 weeks of pregnancy. Data on fetal body weight and biparietal diameter (BPD) indicating fetal development were collected at 33–35 weeks to observe the longitudinal effects of mothers’ emotions on fetal development.

### 2.4. Variables

#### 2.4.1. Independent Variables

Structured questionnaires were used for this study. Socio-demographic information was collected: household income, maternal age, gestational age, and parental occupation status.

##### Maternal Depression

Prenatal depression was measured using the Edinburg Postnatal Depression Scale (EPDS) developed by Cox, Holden, and Sagovsky [10], which was adapted by Yeo [11]. In particular, EPDS has been developed to screen postpartum depression, and now is widely used to measure prenatal depression. It is composed of 10 simple items, is easy to conduct and interpret, and has the advantage of good reliability and validity regardless of the cultural context. The descriptive self-rating scale has a score ranging from 0 to 30 on a 4-point (0 = no depression; 3 = high depression) Likert scale. A score of 12 or less indicates no risk for depression, while 13 or more is considered to indicate a risk for depression. Internal reliability in this study was 0.765.

##### Maternal Stress

Ahn’s maternal stress scale was used [12], which consists of 9 items about stress regarding fetal health and child care after birth, 11 about stress for the mother herself, and 6 about stress related to spouse. The self-reported 5-point Likert scale (1 = low stress to 5 = high stress) had a score ranging from 26 to 130 points. The higher the score, the higher is the perceived stress. Internal reliability in this study was 0.905.

##### Maternal Emotions

The Positive and Negative Affect Schedule (PANAS) developed by Watson [13] and adapted by Park [14] was used in the study. Positive and negative emotions are both evaluated with 10 items, each on a 5-point scale (0 = feel the emotion weakly to 4 = feel the emotion presented strongly). The higher the total positive and negative emotional scores, the higher is the intensity of the emotion. Internal reliability in this study was 0.871 for the Positive emotion subscale and 0.807 for the Negative emotion subscale.

##### Maternal-Fetal Attachment

Cranley, a researcher who created the theoretical construct of MFA, defined it as “the extent to which women engage in behaviors that represent an affiliation and interaction with their unborn child.” Cranley developed the first Maternal Fetal Attachment Scale [15], which was revised by Lee [16] and used in the study. The questionnaire consists of 16 items regarding expectations and efforts for the health of the baby that are considered appropriate for measuring attachment at the beginning and end of pregnancy. The higher the total score, the higher the MFA. It is composed of four subscales: (i) anticipation of interaction with the baby; (ii) interaction with the fetus; (iii) giving of self; and (iv) choice of name. Internal reliability in this study was 0.798.

##### Serum Cortisol Level

Cortisol is a steroid hormone released by the adrenal glands and involved in stress and the fight-or-flight reaction. Adults have normal range of cortisol levels of 5 to 25 μg/dL in the morning to mid-day, but it increases during pregnancy. In the second trimester of pregnancy (13~26 weeks), cortisol level ranges from 10 to 42 μg/dL, and in the third trimester of pregnancy (27~40 weeks), it ranges from 12 to 50 μg/dL [17]. All venous blood samples were collected from the mothers from 9 a.m. to mid-day on the day of the clinical examinations and processed at the laboratory of the hospital. Total cortisol in serum was determined by radio-immunoassay and defined in μg/dL. No systematic differences in the time that the blood samples were collected were noted between participants.

#### 2.4.2. Dependent Variables

Fetal weight and BPD assessed by ultrasound have been used successfully to assess fetal development [18]. Ultrasonography was performed at each time point: 16–20 weeks of pregnancy and 33–35 weeks of pregnancy by trained gynecologists. Fetal weight and BPD were measured accurately with an abdominal probe and were recorded by gynecologists and registered nurses. 

#### 2.4.3. Covariates

Patient characteristics associated with fetal development were regarded as covariates. Two associated factors were considered as confounders of outcome measure based on prior studies: gestational age and maternal weight gain during pregnancy [9]. 

### 2.5. Data Analysis

To describe the characteristics of the study population and key variables, frequency, means, and standard deviation were obtained. In order to analyze the correlation between maternal emotions and fetal development according to the aims of research, participants with more than 150% of median income were classified as the high-income group (≥4,220,000 Korean Won per month; approximately 3793 USD per month) and 50% to less than 150% as the middle-income group (1,400,000 < X < 4,220,000 Korean Won per month). This division was based on the median income of 2017 for a family of two as set by the Ministry of Health and Welfare, South Korea [19]. First, a *t*-test was performed to compare differences between the two groups at each time point. Second, correlation analysis of maternal emotional status at 16–20 weeks of pregnancy (T1) and fetal development at 33–35 weeks of pregnancy (T2) was performed for each group. Last, in order to examine the factors associated with fetal development (fetal body weight) at 33–35 weeks of pregnancy, we used multiple linear regression analysis. In selecting variables, we used the forced enter method for the variables of cortisol, mother’s occupation status, maternal weight gain, and gestational age based on the literature review. We entered the income level variable not only as an interaction term with the mother’s emotional variable but also as an independent variable to examine the influence on fetal development. The final model was fitted by comparing the *p*-value of the variables and explanatory power of each model. After choosing the variable, multicollinearity was diagnosed using the variance inflation factor value. A bootstrapping method was used to obtain a valid 95% confidence interval and standard error. The missing value of household income (7 missing value), BPD (1 missing value), and cortisol (4 missing value) measured at 33–35 weeks were imputed by regression analysis based on parental occupation, fetal weight, and gestational age, and cortisol levels measured at 16–20 weeks, respectively. Boxplot, Cook’s distance, and the standardized residual value were checked and three outliers over the 2.58 standardized residual data were removed to adjust skewed distribution. Confidence intervals (CI) and P values (smaller than 0.05) were used for assessing the significance.

## 3. Results

As Table 1 reveals, the average age of mothers was 29.74 years (SD = 3.94) in the middle-income group, and 30.24 years (SD = 3.54) in the high-income group for which the proportion of mothers with jobs was 87.8%—higher than that of the middle-income group (37.4%). Of pregnant women, 59% were under the 150% median income (middle-income group), and 41% were above it (high-income group). The average maternal weights were 59.76 kg and 69.79 kg at 16–20 weeks and 33–35 weeks of pregnancy, respectively. Cortisol was in the normal range of 10–42 µg/dL in 16–20 weeks of pregnancy (12.66, middle-income group; 12.18, high-income group), and the mothers’ mean cortisol levels were in the normal range of 12 to 50 µg/dL during 33–35 weeks of pregnancy (24.53 µg/dL; middle-income group, 23.44 µg/dL; high-income group) [17]. According to World Health Organization’s longitudinal data on fetal development, mean fetal weight and BPD in this study fall within the normal range of the relevant gestational age [20]. T-tests and chi-square tests did not show any significant differences in the general characteristics of the two groups except mother’s employment status.

Table 2 presents the results of *t*-tests on maternal emotional status and fetal development by group. The groups were significantly different in the measures of depression (*t* = 3.630, *p* < 0.01) and stress (*t* = 2.323, *p* < 0.05) at 16–20 weeks of pregnancy. There was no statistically significant difference between the two groups in the mean of fetal weight, BPD, or emotional state at 33–35 weeks of pregnancy.

The results of the correlational analysis between maternal emotion and fetal development by household income group are presented in Table 3. What stands out in the table is that only the middle-income group showed significant positive correlations with fetal weight and BPD at 33–35 weeks of pregnancy and the mother’s negative emotion and MFA level at 16–20 weeks. Increased negative emotions of mothers corresponded to lower fetal weight (r = −0.256, *p* < 0.05) and BPD (r = −0.380, *p* < 0.01). Moreover, higher MFA level correlated to higher fetal weight (r = 0.241, *p* < 0.005) and BPD (r = 0.286, *p* < 0.05). In the middle-income group, cortisol levels at 33–35 weeks were negatively correlated with fetal body weights at 33–35 weeks (r = −0.256, *p* < 0.05). Although statistical significance was confirmed only in the high-income group, negative emotions and MFA showed a negative correlation (r = −0. 329, *p* < 0.05).

The results of the multiple regression analysis of fetal development are presented in Table 4. We considered gestational age and maternal weight gain during pregnancy as covariates based on literature review. In the process of constructing regression model, the household income variable was included as an independent variable and as interaction terms with the mother’s emotional variables. However, the interaction terms were not statistically significant and lower the explanatory power of model. The final regression model was confirmed which has the highest explanatory power focusing on the important variable in the review of precedent research. Therefore, the final model consisted of variables of covariates (gestational age and maternal weight gain), household income, mother’s occupation status, cortisol level at third trimester, MFA and mother’s positive emotion at second trimester. F-test results of the regression model was significant at the significance level of < 0.001. The value of Durbin-Watson was close to 2 (Durbin-Watson = 1.72), so we assumed that there were no correlations between the residuals. In the regression model, the MFA was statistically significant (*b* = 7.567, *p* < 0.01) and the regression coefficient was positive. Therefore, we can assume that the MFA would have a positive association on fetal development. Neither the cortisol level, the mother’s job status, nor maternal positive emotion were associated with fetal development. All VIF values were less than 10 and there was no problem of multi-collinearity.

## 4. Discussion

The discourse on human rights refers to the discussion about how to preserve and protect certain minimal rights necessary for all individuals to enjoy a high quality of life. More recently, the right to health has been included in quality of life measurements from a maximalist point of view expanding human rights beyond food, water, and housing. The right to health emerged from efforts to solve health problems such as poverty, starvation, and death due to diseases by providing medical and health care systems to satisfy the minimum conditions under which humans can live [21]. Health as a right cannot be attained only by meeting the conditions of survival, but it is a problem that requires political, cultural, and community approaches rather than individual efforts. Individual health is influenced not only by the health care system but also by various complex factors such as lifestyle, education, and socioeconomic inequality. Prior studies have noted the importance of maternal mental health during pregnancy, which can be affected by education, social support, violence, and prenatal care, depending on income level [5]. Therefore, this study aimed to identify whether mothers’ emotions and MFA are associated with the development of the fetus during pregnancy by income level.

Consistent with the literature review, in the middle-income group, depression and stress were found to be statistically significantly higher than that in the high-income group, and positive emotions and MFA were lower than that in the high-income group even though there was no statistical significance. Depressive mood has recently been found to be similar between pregnant women and similarly aged non-pregnant women [22]. However, stress levels and depression during pregnancy are associated with premature birth, low birth weight, and intrauterine growth retardation (IUGR) [23], and such diagnoses may also be associated with an infant’s allergic response after childbirth [24]. Therefore, more preventive management is needed. Depression, anxiety, and stress have shown higher prevalence rates in developing countries, suggesting that socioeconomic levels may affect pregnant women’s emotions [6].

A fetus’ attachment with the mother is its first human relationship. Rubin [25] notes that the immediate attachment of newborns with mothers is the result of their previous attachment during pregnancy, and pregnancy is the beginning of the process of acquiring the role of the mother. Recent systematic review of the literature suggests that depression, anxiety, and substance abuse are factors that reduce fetal attachment and that factors associated with higher socioeconomic status, such as access to appropriate prenatal care or stable family relationships, are highly associated with MFA [26]. In this study, although statistical significance was confirmed only in the high-income group, negative emotions and MFA were negatively correlated. Previous studies have also shown that MFA levels decrease as the mother has negative emotions, but there were no significant correlations between MFA, abortion, prematurity, and IUGR. Previous research also showed no statistically significant correlation with MFA and abortion, prematurity, or IUGR [27].

In a study by Kwon and Bang [28], the relationship between MFA level and fetal body weight was statistically insignificant. However, it is interesting to note that the correlations between maternal attachment and negative emotions were found to be associated with fetal development in middle-income groups. A possible explanation for this might be that adequate maternal care and routine screening are possible in the case of the high-income groups because of the abundance of available resources, regardless of the mother’s emotions. However, in the case of the middle-income group, the mothers’ emotions influenced the mother’s self-care and behavior of caring for the fetus during pregnancy more than that in the high-income group. As stated above, increased negative emotions among pregnant women may lead to lower MFA and result in unhealthy behaviors, such as inappropriate nutrition, smoking, and lack of regular checkups.

One unanticipated finding was that there was no statistically significant correlation between depression and fetal weight in this study, but Diego et al. [29] found a correlation between intrauterine growth rate and depression. The theory that explains this is that cortisol levels increase in depressed mothers and 10–20% of the cortisol passes through the placenta to stimulate the fetal hypothalamic-pituitary-adrenal axis and dysregulate the fetal autonomic nervous system, resulting in high calorie consumption. In the mid-trimester, high cortisol levels may reduce intrauterine arterial flow and be associated with spontaneous preterm birth and fetal growth retardation [30,31]. In general, cortisol in pregnant women is higher than that in non-pregnant adult women under normal conditions and increases toward the end of pregnancy [8,32]. Previous studies have reported a negative correlation between cortisol and fetal development [8,29]. In the present study, we found that the direction of the correlational relationship was negative but statistically insignificant.

Inequality in early life can lead to cumulative and latent health effects. Therefore, it is necessary to analyze the longitudinal effects by comparing previous living conditions in order to explain the present health differences considering the lagged exposure [33]. Based on this logic, we conducted a multiple regression analysis on fetal development during the third trimester of pregnancy through mothers’ emotional status in the second trimester of pregnancy as well as their occupational status. Income levels were found to be statistically insignificant in this regression model. Participants described their household income in units of 500,000 won rather than with continuous data. Thus, the self-response data of household income might not have been clear. Therefore, further studies may be needed that measure household income more accurately, such as by using health insurance fees. In Scotland and Spain, the prevalence of IUGR was higher when the mother held a blue-collar job than a white-collar job [32,33]. In several epidemiological studies, strenuous physical labor has been reported to cause premature birth or IUGR, but “strenuous” refers to such work as prolonged standing. Thus, as with the findings of this study, results suggest that the presence or absence of an occupation for mothers may be less likely to contribute to IUGR. Further research on the effects of occupations is required to examine the type of work and the working environment. There are very few studies on MFA as a predictor of fetal development. Hompes et al. [8], with 91 pregnant women who were recruited from a prenatal care clinic, confirmed that MFA had a significant influence on fetal weight and head circumference in the third trimester of pregnancy, but the reason for the negative effects that they found remains unclear. In this study, MFA was a significant variable in the multivariate model, regardless of income level.

Another controversy about health inequity is the notion that this research leads to another prejudiced causal relationship associated with the social class. However, we not only want to know the basic health differences, but also learn about social factors and their role in systematically influencing health.

These findings may help us to understand that the emotions of pregnant women may be different according to income level, and their effects on fetal development may also be different. Therefore, it can be suggested that emotional management may be more important in low-income mothers than high-income ones. However, we conducted the study with participants from a single hospital and there were no low-income mothers in the sample. Therefore, the size and representativeness of the sample may somewhat limit these findings. To develop a full picture of how mothers’ emotions and fetal attachment are associated with fetal development, additional studies are needed with larger and more diverse populations, including low-income mothers.

## 5. Conclusions

The aim of the present study was to identify how mothers’ emotion and MFA are associated with the development of the fetus during pregnancy by income level. The results of this study show that middle-income mothers’ depression and stress levels were found to be statistically significantly lower than that of the high-income group, and positive emotions and fetal attachment were lower than that in the high-income group, though there was no statistical significance in the latter case. The second major finding was that the middle-income group showed a significantly positive correlation between fetal weight and BPD at 33–35 weeks of pregnancy and between the mother’s negative emotions and MFA level at 16–20 weeks. Multiple regression analysis revealed that fetal development can be increased in the mothers who have more MFA. Moreover, this study strengthens the idea that the emotions of pregnant women may be different according to income level, and their effects on fetal development may also be different. Therefore, it can be suggested that emotional management can be more important in low- than high-income mothers. A further study with a larger sample and diverse settings, including low-income mothers, is therefore suggested.

## Figures and Tables

**Table 1 ijerph-16-02065-t001:** Descriptive statistics of participants’ characteristics (*n* = 121).

Characteristics	Categories	N (%) or M ± SD		
		Middle-Income Group (*n* = 72)	High-Income Group (*n* = 49)	*t* or *χ*^2^
Maternal age	-	29.74 ± 3.94	30.24 ± 3.54	−0.73
Mother’s occupational Status	Employed	25 (34.7)	43 (87.8)	33.313 **
	Unemployed	47 (65.3)	6 (12.2)	
Father’s occupational Status	Employed	71 (98.6)	49 (100)	0.686
	Unemployed	1 (1.4)	0 (0)	
Gestational age (day)	16–20 weeks	113.28 ± 5.42	112.78 ± 6.21	−0.179
	33–35 weeks	240.57 ± 8.23	238.98 ± 7.78	1.07
Maternal weight (kg)	16–20 weeks	60.31 ± 9.15	58.96 ± 8.82	0.806
	33–35 weeks	70.49 ± 9.25	68.35 ± 9.78	1.22
Maternal weight gain (kg)	(33–35 weeks)–(16–20 weeks)	10.18 ± 3.46	9.39 ± 3.43	1.24
Cortisol (µg/dL)	16–20 weeks	12.66 ± 4.73	12.18 ± 3.97	0.58
	33–35 weeks	24.53 ± 6.18	23.44 ± 7.42	0.875

Notes: ** *p* < 0.01.

**Table 2 ijerph-16-02065-t002:** Differences between middle-income group and high-income group on maternal emotional status and fetal development (*n* = 121).

Variable	Middle-Income Group (*n* = 72)	High-Income Group (*n* = 49)	
	M ± SD	M ± SD	*t*
T1: 16–20 Weeks			
Positive emotion	19.08 ± 7.26	21.1 ± 6.55	−1.560
Negative emotion	6.59 ± 5.98	5.42 ± 4.70	1.199
Depression	7.68 ± 3.96	5.12 ± 3.55	3.630 **
Maternal-fetal attachment	60.2 ± 8.48	62.4 ± 8.17	−1.447
Stress	70.56 ± 13.2	64.3 ± 15.8	2.323 *
Fetal weight (g)	161.03 ± 37.37	161.67 ± 36.43	−0.094
Biparietal diameter (cm)	3.49 ± 0.34	3.45 ± 0.35	0.728
T2: 33–35 Weeks			
Positive emotion	19.72 ± 6.65	20.37 ± 7.3	−0.504
Negative emotion	8.88 ± 7.28	8.9 ± 7.31	−0.017
Depression	6.58 ± 4.14	5.73 ± 3.66	1.159
Maternal-fetal attachment	62.19 ± 8.47	63.27 ± 7.77	−0.706
Stress	68.9 ± 16.69	63.24 ± 19.36	1.715
Fetal weight (g)	2404.89 ± 317.26	2347.84 ± 365.91	0.912
Biparietal diameter (cm)	8.76 ± 0.34	8.68 ± 0.44	1.035

Notes: * *p* < 0.05, ** *p* < 0.01.

**Table 3 ijerph-16-02065-t003:** Correlations among maternal emotional status and fetal development.

	1	2	3	4	5	6	7	8	9	10	11	12	13	14	15	16	17
1. Maternal Weight Gain	1	0.068	−0.179	−0.195	0.049	0.038	−0.029	−0.134	−0.168	0.036	0.024	−0.164	−0.051	0.155	−0.006	0.338 *	0.248
2. Cortisol1	−0.046	1	−0.259	0.001	0.124	0.089	0.068	0.234	0.253	0.243	−0.003	−0.122	0.089	−0.103	−0.019	−0.061	0.066
3. Positive emotion1	−0.038	0.017	1	−0.066	−0.270	0.271	−0.353*	−0.140	−0.013	0.188	0.475 **	0.105	−0.145	0.342 *	−0.315 *	0.057	−0.020
4. Negative emotion1	−0.010	−0.140	0.000	1	0.483 **	−0.329 *	0.391 **	0.032	0.025	−0.036	−0.111	0.413 **	0.062	−0.226	0.313*	0.049	0.096
5. Depression1	−0.105	−0.104	−0.368 **	0.494 **	1	−0.186	0.608 **	0.044	−0.033	−0.173	−0.140	0.327 *	0.466 **	0.105	0.533 **	−0.011	0.084
6. Maternal-Fetal Attachment1	−0.096	0.106	0.167	−0.058	0.078	1	−0.114	0.136	0.048	0.222	0.367 **	−0.229	−0.123	0.555 **	−0.089	−0.010	0.193
7. Stress1	0.100	−0.111	−0.400 **	0.345 **	0.484 **	0.110	1	0.202	0.107	−0.135	−0.320 *	0.496 **	0.485 **	−0.101	0.763 **	−0.015	0.036
8. Fetal Weight1	−0.124	0.316 **	0.057	−0.218	−0.123	0.169	−0.211	1	0.871 **	−0.040	−0.205	0.107	0.151	−0.159	0.174	−0.081	0.016
9. Biparietal Diameter1	−0.149	0.218	0.028	−0.270 *	−0.106	0.085	−0.189	0.889 **	1	0.097	−0.167	0.119	0.160	−0.191	0.111	−0.112	0.017
10. Cortisol2	0.121	0.416 **	−0.022	0.123	0.099	−0.037	−0.009	0.052	−0.030	1	0.243	−0.148	−0.055	0.184	−0.230	−0.100	0.045
11. Positive Emotion2	−0.101	−0.051	0.318**	0.111	−0.041	0.175	0.012	−0.145	−0.133	−0.146	1	−0.023	−0.228	0.571 **	−0.528 **	0.117	0.274
12. Negative Emotion2	0.041	−0.146	−0.267 *	0.295 *	0.424 **	0.208	0.428 **	−0.088	−0.084	0.027	0.114	1	0.514 **	−0.128	0.571 **	0.028	−0.027
13. Depression2	0.108	−0.105	−0.093	0.391 **	0.422 **	0.193	0.329 **	−0.099	−0.116	0.189	−0.298 *	0.600 **	1	−0.082	0.582 **	0.146	0.130
14. Maternal-Fetal Attachment2	−0.084	0.049	0.124	0.076	−0.050	0.432 **	0.060	0.071	0.065	0.014	0.518 **	−0.035	−0.105	1	−0.231	−0.060	0.192
15. Stress2	0.209	0.037	−0.251 *	0.124	0.273 *	0.079	0.513 **	0.011	0.071	0.118	−0.238 *	0.484 **	0.424 **	−0.177	1	0.071	0.036
16. Fetal Weight2	0.167	−0.074	−0.114	−0.256 *	−0.221	0.241 *	−0.064	−0.080	−0.138	−0.256 *	−0.039	0.034	−0.050	0.030	−0.087	1	0.659 **
17. Biparietal Diameter2	0.037	0.045	−0.104	−0.380 **	−0.178	0.286 *	−0.135	0.036	0.050	−0.122	−0.077	0.072	0.021	−0.011	−0.012	0.564 **	1

Note 1. The number next to the variable indicates the time of the investigation. 1 = 16–20 weeks, 2 = 33–35 weeks. Note 2. Correlation values above the diagonal line represent the high-income group (*n =* 49), and those below the diagonal line represent the data of the middle-income group (*n =* 72). * *p* < 0.05, ** *p* < 0.01.

**Table 4 ijerph-16-02065-t004:** Linear model summary for maternal emotional status, cortisol level, mother’s job status, and covariates on fetal weight.

	Dependent Variables	Fetal Weight (g)					
		*B*	*SE B*	β	*p*	*95% CI*	
Independent Variables						*LL*	*UL*
Intercepts		−5049.039	653.763	-	<0.001	−6344.262	−3753.816
Household income		−0.343	0.240	−0.103	0.156	−0.818	0.133
Cortisol(33–35 weeks)		−4.700	3.066	−0.093	0.128	−10.774	1.374
Mother’s occupation status		96.196	49.462	0.142	0.054	−1.798	194.190
Positive emotion (16–20 weeks)		−3.033	2.969	−0.063	0.309	−8.915	2.848
Maternal-fetal attachment (16–20 weeks)		7.567	2.448	0.188	0.003**	2.716	12.417

Note 1. *R*^2^ = 0.608, *Adjusted*
*R*^2^ = 0.584 (F = 25.04, *p* < 0.01), Durbin-Watson = 1.72. Note 2. SE = Standard error; CI = Confidence interval; LL = Lower limit; UL = Upper limit. Note 3. 95% bias corrected and accelerated confidence intervals reported. Confidence intervals and standard errors based on 1000 bootstrap samples. Note 4. Gestational age (days) and maternal weight gain during pregnancy are covariates in model (*p* < 0.01).

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
