# Peer review of "Do the Emotions of Middle-Income Mothers Affect Fetal Development More Than Those of High-Income Mothers?—The Association between Maternal Emotion and Fetal Development"

_ijerph, 2019, doi:10.3390/ijerph16112065_

Round 1

Reviewer 1 Report

Thanks for your effort in revising your manuscript.
This study has been well modified in accordance with the opinion of the reviewers.
The revised manuscripts are well organized to the extent that they can be published in the IJERPH.

Author Response

Please find authors' reply attached.

Reviewer 2 Report

Dear authors,

Thank you for the opportunity to review your manuscript. This manuscript is much improved. I have a few remaining comments.

(1)   Starting on line 72: power calculation is not important for the manuscript. This information is useful prior to starting the study. Your sample size is what it is. You cannot change it at this point.

(2)   Table 1 has Father’s occupation at Yes/No instead of Employed and Unemployed

(3)   Table 2 has formatting problems.

(4)   Is the low-income group if there is a middle and a high-income group?

(5)   Step-wise regression to finalize the model is not needed when covariate selection is based off subject matter expertise. I would remove the element of step-wise selection altogether.

Dichotomizing income and then presenting results by that dichomtomization does not utilize the data from the income variable. Instead of a specific value for each participant you have two values and only two values. If this is a key point of your paper (modification by income), you should really present results with income as a continuous interaction variable.

Author Response

Please find authors' reply attached.

This manuscript is a resubmission of an earlier submission. The following is a list of the peer review reports and author responses from that submission.

Round 1

Reviewer 1 Report

Thanks for your effort for working on this paper.

The significance of this research is that it provides an empirical basis for emotional management may be more important in low-income mothers than in high-income ones.

The manuscripts are well organized to the extent that they can be published in the International Journal of Environmental Research and Public Health. But minor revision needed.

- Please mention the method of calculating the sample size.

- Information on the reliability of scales is not stated.

- Page 5, line175: “Notes: * p<0.05, **p<0.01.” There are no “*” marks in the Table1

- Page 8, Table 4; Intercepts: p value “0.000**” should be corrected to p<.001”

Author Response

Thank you for inviting us to submit a revised draft of our manuscript entitled, “Do the emotions of middle-income mothers affect fetal development more than those of high-income mothers?: The association between maternal emotion and fetal development” to International Journal of Environmental Research and Public Health. We also appreciate the time and effort you and each of the reviewers have dedicated to providing insightful feedback on ways to strengthen our paper. Thus, it is with great pleasure that we resubmit our article for further consideration. We have incorporated changes that reflect the detailed suggestions you have graciously provided. We also hope that our edits and responses provided below satisfactorily address all the issues and concerns you and the reviewers have noted.

To facilitate your review of our revisions, attached file is point-by-point responses to the questions and comments delivered in your letter dated March 13, 2019.

Reviewer 2 Report

Dear authors: Dasom Kim, Insook Lee, Kyung-Sook Bang, Sungjae Kim, and Yunjeong Yi.

I thank you for the opportunity to read your article entitled, "Do the emotions of middle-class mothers affect fetal development more than those of upper-class mothers?: The association between maternal emotion and fetal maturity". This manuscript has merit and I look forward to seeing its progression. That said, I have a number of major and minor concerns that prevent it from being publication ready at this time.

Major concerns/questions

(1) The manuscript use strong causal language (X affects Y) when the results cannot speak to this. I would not include such statements in the results and abstract sections.

(2) English language use is confusing in a number of places. Word choice can be improved. For example, what does nursing refer to breastfeeding or the practice of being a nurse? Also participants should not be referred to as samples. Use of moderating vs. modifying. A careful review of the writing needs to be done

(3) There are only stratified results present for Table 4. What are the results not stratified by income group? Stratification assumes effect measure modification by all variables in the model. Why not just include an interaction term in the linear regression model?

(4) Interpretation of Table 4 results is inadequate. Please provide a clear interpretation of the primary outcome and exposure relation.

Minor concerns/questions

(1) Why was this research funded by the Forest Service?.

(2) Bootstrapping is described as a significance testing method. This is incorrect.

(3) Unless your objective is prediction (not understanding etiology) covariate selection should not be determined on statistical significance.

(4) The formatting needs to be improved for some of the tables and other parts of the manuscript

I have provided a pdf copy of some handwritten edits and comments as well. Please look through those. Thank you for sharing your work. I look forward to seeing this research progress.

Author Response

(The authors gave the same response as above.)

Round 2

Reviewer 2 Report

Dear authors,

Thank you for your work to improve your manuscript. I still have a number of concerns.

(1) the following explanation of the bootstrapping method is inadequate/incorrect. 

"To improve the symmetry of the data distribution and precision of the data 178 analysis, we used bootstrapping method by resampling 1000 data from original sample."

This makes the method sound as if you have inflated your sample size artificially for better precision.

Bootstrapping, as it is described in the footnotes of one of the tables is a method of random draws from the original sample for generating valid 95% CIs. This seems correct but the above section of the methods does not indicate this.

(2) I am still confused about why the Forest Service would fund a pregnancy related study. Would you mind explaining this better.

(3) There is still causal language in the results (see line 225: "Therefore, we can assume that the MFA would have influence on fetal development."). I would recommend providing interpretations in the results section that do not focus solely on statistical significance. Much of the results seem to just focus on whether or not something is statistically significant. For example lines 225-226: "However, cortisol and mothers' job status and maternal positive emotion were not significant in both groups." could be written as 'Among middle income mothers, cortisol level was no associated with fetal growth.'

(4) Since the middle income group includes low income mothers, should the group be named differently?

(5) You model selection is based on statistical significance when you use stepwise selection to build the model. Therefore the model does not appear to be built as you specified in your reply. It appears to be model built more for prediction (see also title of Table 4) rather than understanding causal relations.

(6) If as you said in your reply "The interaction terms between the variables were not significant and did not increase the explanatory power of the model. Therefore, we have eliminated the interaction terms." Why would you have the results stratified by a dichotomous income group?

Best,

A.R.